# Level of Combined Estrogen and Progesterone Receptor Expression Determines the Eligibility for Adjuvant Endocrine Therapy in Breast Cancer Patients

**DOI:** 10.3390/cancers13195007

**Published:** 2021-10-06

**Authors:** Jee Hyun Ahn, Soon Bo Choi, Jung Min Park, Jee Ye Kim, Hyung Seok Park, Seung Il Kim, Byeong-Woo Park, Seho Park

**Affiliations:** Division of Breast Surgery, Department of Surgery, College of Medicine, Yonsei University, Seoul 03722, Korea; jhahn35@yuhs.ac (J.H.A.); csbnhi@yuhs.ac (S.B.C.); minnyang26@yuhs.ac (J.M.P.); jeeye0531@yuhs.ac (J.Y.K.); imgenius@yuhs.ac (H.S.P.); skim@yuhs.ac (S.I.K.); bwpark@yuhs.ac (B.-W.P.)

**Keywords:** estrogen receptor, progesterone receptor, gene expression, endocrine therapy, breast cancer, neoplasms by histologic type, survival analysis

## Abstract

**Simple Summary:**

Patients whose breast cancers express low levels of hormone receptor (HR) could be eligible for adjuvant endocrine therapy; however, limited data are available to support this notion. Our retrospective study investigated the characteristics and survival of 6042 breast cancer patients according to four HR groups of combined estrogen and progesterone receptor expression. HR expression levels were prognostic for its recurrence and death of patients with breast cancer. Patients whose tumors expressed high levels of a single HR had the worst survival outcomes, and their risk of death continuously increased even after the 10-year follow-up. Endocrine therapy had a significant benefit for those whose tumors expressed high HR levels and a favorable tendency for patients with tumors expressing low HR levels. We established the value of HR expression level as a prognostic factor and the possible benefit of endocrine therapy for patients whose breast tumors expressed low HR levels.

**Abstract:**

Hormone receptor (HR)-positive breast cancer has a heterogeneous pattern according to the level of receptor expression. Patients whose breast cancers express low levels of estrogen receptor (ER) or progesterone receptor (PgR) may be eligible for adjuvant endocrine therapy, but limited data are available to support this notion. We aimed to determine whether HR expression level is related to prognosis. Tumors from 6042 patients with breast cancer were retrospectively analyzed for combined HR levels of ER and PgR. Low expression was defined as ER 1–10% and PgR 1–20%. Four HR groups were identified by combining ER and PgR expression levels. Patients whose tumors expressed high levels of a single receptor showed the worst survival outcomes, and their risk continuously increased even after the 10-year follow-up. Endocrine therapy had a significant benefit for patients whose tumors expressed high HR levels and a favorable tendency for patients with tumors expressing low HR levels. We established the possible benefit of endocrine therapy for patients whose breast tumors expressed low HR levels. Thus, HR level was a prognostic factor and might be a determinant of extended therapy, especially for patients with high expression of a single receptor.

## 1. Introduction

The incidence of breast cancer, including hormone receptor (HR)-positive tumors, has been increasing globally over the past few decades [1,2]. The estrogen receptor (ER) and the progesterone receptor (PgR) are among the receptors of the sex steroid hormones. ER and PgR are a few of the essential biomarkers related to the clinical patterns, treatment decision, and prognosis of breast cancer. ER is expressed in approximately 70–80% of patients with breast cancer who are recommended endocrine therapy [3]. With the establishment of immunohistochemistry (IHC) staining as an effective method for determining HR expression level in breast cancer [4], the predictive value of HR expression for determining the survival outcome of patients has been actively explored in several studies [5,6]. 

According to the guidelines of the American Society of Clinical Oncology (ASCO) 2010, the positive status of ER or PgR, which is defined as the presence of at least 1% stained cancer nuclei of ER or PgR, is an indication for subsequent endocrine treatment [7]. However, breast cancer is well-known as a heterogeneous disease. ER and PgR are expressed to varying degrees, which results in differences in the prognosis and/or responsiveness of breast cancer to endocrine treatment [8,9,10]. 

The lower the ER or PgR expression levels are, the closer the clinical patterns are to HR-negative breast cancer. In an earlier study on ER expression using IHC staining in 1999, patients with an Allred score of two had similar clinical features to patients with negative IHC staining and showed a poorer prognosis than that in patients with a high Allred score [4]. The Allred score ranges from 0 to 8 and is determined by the proportion and intensity of expression of the stained tumor nuclei. The results suggested that the clinical behavior of patients with the subtype of ER-positive breast cancer having tumors with low ER expression was different from the clinical behavior of patients with high ER expression in the breast tumors. 

Breast cancers expressing higher than 10% of ER and higher than 20% of PgR show an optimized tendency to be considered as typical HR-positive breast cancers [10,11,12]. In 2012, the ASCO guidelines announced that 1–9% ER-positive breast cancers detected using IHC had similar patterns at the molecular level to ER-negative breast cancer [13]. In the St. Gallen consensus released the following year, it was defined that the best cutoff for PgR positivity with characteristics of HR-positive breast cancer was at least 20% [14]. Most HR-positive breast cancers are known to progress slowly, respond well to endocrine therapy that inhibits receptor-mediated signaling pathways, and the affected patients generally have favorable survival outcomes. However, for patients with breast tumors expressing low levels of HR, categorically defined as 1–9% of ER or 1–20% of PgR, it is recommended to add more aggressive treatments such as chemotherapy, with endocrine therapy, for the oncologic safety of the patients [15].

Despite the global consensus, many studies suggest other thresholds for determining the survival outcomes of patients with breast tumors expressing low ER or PgR levels. Among these are tumors expressing 1–5% [16] and 1–10% [7,17,18] of ER and, similarly, 1–5% [16] and 1–10% [17] of PgR. Several researchers focused on the level of an ER/PgR pair expression, that is, 1–5% [16], or 1–10% [17] of both ER and PgR. However, these studies were performed with ER or PgR alone or were limited to the lower limit of an ER/PgR pair expression. Although the mechanism of the occurrence of breast cancers with ER/PgR +/− and −/+ is unclear, they occur at a rate of 2–5% of all the breast cancers [10,18,19,20]. However, because the binary definition of ER or PgR positivity, based on a different cutoff level in previous studies, it is questionable whether a low level of ER- or PgR-positive breast cancers could be categorized as a high level of HR-positive tumors, which typically showed favorable biological features. Therefore, there is a strong need to analyze the entire spectrum of ER and PgR expression levels for breast tumors.

This study aimed to investigate the characteristics and survival outcomes of patients with breast cancer based on the HR expression level of their tumors. We determined negative, low, and high levels each of ER and PgR expression according to the previous consensus, and then considered the combined effect of the HR expression level. We evaluated whether the HR expression level is sufficiently related to prognosis and determined the patient population that could benefit from adjuvant endocrine therapy.

## 2. Materials and Methods

This study was approved by the institutional review board of the Severance Hospital Yonsei University Health System (IRB 4-2021-0079), and the need for informed consent was waived due to the retrospective nature of this research.

### 2.1. Patients

From the institutional database, we retrospectively extracted the anonymized data of 6042 patients with breast cancer who underwent definite breast surgery at the Severance Hospital, Seoul, Korea from January 1997 to December 2014. Patients who received neoadjuvant chemotherapy, did not undergo definite breast surgery, had metastatic disease at the time of diagnosis, or were diagnosed with carcinoma in situ, sarcomas, lymphomas, and phyllodes tumor of the breast were excluded from the study (Figure 1). In addition, subjects with unavailable data for pathologic results related to cancer staging or HR status were also excluded from the study.

The patient database contains information on the patients’ demographics, pathology, adjuvant treatments, disease relapse, and death event occurrence. Information about the patient’s body mass index (BMI), histologic grade of the tumor, and human epidermal growth factor receptor 2 (HER2) overexpression were collected, if available. The TNM staging of breast cancer was determined based on the staging system proposed by the 7th American Joint Committee on Cancer system [21]. Endocrine therapy included medications using selective ER modulators or aromatase inhibitors, medical ovarian function suppression, and surgical bilateral oophorectomy. Intention-to-treat analysis was applied, and the duration of endocrine therapy was not considered. Adjuvant chemotherapy also followed prior rules, and anti-HER2 treatment was not considered because it had not been used during a major part of the study period.

### 2.2. IHC Staining

The levels of ER and PgR expression were determined using the proportional degree of IHC staining. ER expression levels were categorized as follows, according to the recommendations of the ASCO/College of American Pathologists: <1% was classified as negative, 1–10% was classified as low, and ≥10% was classified as high (Figure 2). Furthermore, PgR expression levels were also classified as negative (<1%), low (1–20%), and high (≥20%). Finally, HR expression levels were grouped by combining the expression levels of ER/PgR (Figure 3). The negative breast cancer group was defined as patients with breast tumors that were both ER- and PgR-negative. The low group comprised patients with breast tumors with low ER and low/negative PgR or negative ER and a low PgR expression level. The single high group consisted of patients whose tumors had only one of the receptors expressed at a high level; that is, ER > 10% and PgR = 0–20% or ER = 0–10% and PgR > 20%. The both high HR group comprised patients whose tumors expressed high ER (>10%) and high PgR (>20%) levels.

HER2 overexpression was evaluated using the Hercep test (Polyclonal, dilution 1:1500; DAKO, Produktionsvej, Glostrup, Denmark), and if necessary, additional fluorescence in situ hybridization (FISH; PathVysion kit, Vysis, Downers Grove, IL, USA, or HER2 inform; Ventana, Tucson, AZ, USA) or silver in situ hybridization (SISH; INFORM HER2 Dual ISH DNA Probe Cocktail Assay, Ventana, Tucson, AZ, USA) was performed. Based on the guidelines of the College of American Pathologists, IHC and ISH results of HER2 expression status were defined as follows: HER2-positive was IHC 3+ or IHC 2+/ISH+, whereas HER2-negative was IHC 0 or IHC 1+ (Figure 4) [22]. Cases with HER2 IHC 2+ but an unavailable ISH test result were labeled as undetermined HER2 (Figure 5).

### 2.3. Statistical Analyses

Analysis of variance was used to compare the average of continuous variables, such as age and BMI. The comparison of categorical variables such as age divided by 50 years, BMI divided by 23 kg/m^2^, tumor size, node metastasis, histologic grade, HR groups, HER2 overexpression, breast surgery, radiation therapy, adjuvant chemotherapy, and endocrine therapy were evaluated using the chi-square (Χ^2^) test. The disease-free survival (DFS) and overall survival (OS) curves were generated using Kaplan–Meier analysis. Events determining DFS were defined as recurrences of local, regional, locoregional, or distant metastasis; contralateral breast cancer; or death. Events determining OS were defined as death from any cause. The follow-up period was calculated from the date of the first breast surgery to the date of the first diagnosis of metachronous breast cancer (>120 days after first breast operation), recurrence, or metastasis or the date of death or the last follow-up. Multivariate Cox regression models were used for the evaluation of the prognostic impact of the variables, calculating hazard ratios. All statistical analyses were carried out using SPSS (version 25.0, IBM Software, IBM, Armonk, NY, USA). A *p*-value less than 0.05 was considered statistically significant.

## 3. Results

### 3.1. Patient Characteristics

A total of 6042 patients with breast cancer were included in the study and divided into four groups. The negative, low, single high, and both high groups were composed of 1533 (25.4%), 229 (3.7%), 1680 (27.8%), and 2600 (43.0%) patients, respectively. The baseline characteristics of the patients are summarized in Table 1. The mean patient age was 50.4, 50.7, 52.4, and 49.1 years for the negative, low, single high, and both high groups, respectively (*p* < 0.001). The average BMI of all the patients was 23.5 kg/m^2^, and there was no significant difference in the BMI between the four HR level groups (*p* = 0.198). 

The most common histologic type of breast cancer was ductal carcinoma. The patients in the negative and low groups had larger T2 to T4 tumors than those patients in the single and both high groups (paired; 44.3% vs. 32.3%, *p* < 0.001). Node metastasis was more frequent among the patients in the single high and both high groups than in the patients of the negative and low groups. The median number of metastatic lymph nodes of the patients in all the four HR groups was two, and the range for the number of metastatic lymph nodes was 1–30 in the negative group, 1–26 in the low group, 1–44 in the single high group, and 1–44 in the both high group. There were more patients with poorly differentiated or overexpressed HER2 tumors in the negative and low groups than those in the single high and both high groups. 

More patients in the negative and low groups were administered adjuvant chemotherapy than those in the single high and both high groups. Cyclophosphamide, methotrexate, and fluorouracil (CMF) or anthracycline-based chemotherapy regimen were more frequently used for patients in the negative and low groups than for those in the single high and both high groups (*p* < 0.001). Taxane-based chemotherapy was adjusted more for the patients in the single high and both high groups than for those in the negative and low groups (paired; 19.2% vs. 14.4%, *p* < 0.001). In summary, 24.5% of the patients in the low group, 4.3% of the patients in the single high group, and 2.5% of the patients in the both high group were not treated with endocrine therapy, which suggested that most of the patients in the HR-positive groups received endocrine therapy (*p* < 0.001).

### 3.2. Survival Outcomes

The average follow-up period was 107 months (range: 0–282 months). We found the recurrence of breast cancer in 299 (19.5%) patients in the negative group, 47 (20.5%) in the low group, 300 (17.9%) in the single high group, and 339 (13.0%) in the both high group. The number of patient deaths that occurred was 220 (14.4%) in the negative group, 33 (14.4%) in the low group, 213 (12.7%) in the single high group, and 229 (8.8%) in the both high group. 

The DFS and OS of the patients, according to the HR expression level in the tumors of the four groups, were analyzed as shown in Figure 6. The HR expression level was significantly associated with long-term prognosis. Within the 10-year follow-up, the patients in the both high group, followed by the single high group, and then the low and negative groups showed better prognosis in terms of both DFS and OS. However, the patients in the single high group showed a continuous decrease in the survival rate, and after 10 years, the prognosis of these patients was the worst among the four groups. At the long-term follow-up, the patients from the both high group followed by the low and negative group had a good prognosis, with no significant difference in the survival outcomes. 

Additional analysis was performed only for the patients in the single high group, based on the type of receptor; 1503 (89.5%) patients in the ER-single high and 177 (10.5%) patients in the PgR-single high groups were considered for the analysis. No significant difference in the DFS and OS was observed between the patients in the ER-single high group and the PgR-single high group (DFS, *p* = 0.935; OS, *p* = 0.750).

After adjusting for factors using multivariate Cox regression analysis, the patients in the single high group had significantly worse DFS and OS than those in the other groups, as shown in Table 2. Traditional prognostic factors such as old age, obesity, large tumor size, and metastasis to axillary lymph node elevated the risks of recurrence and mortality in patients with breast cancer. A low to intermediate histologic grade of the tumors was not identified as an independent risk factor related to the survival outcomes of the patients (DFS, *p* = 0.193; OS, *p* = 0.682). Adjuvant chemotherapy and endocrine therapy of the patients significantly reduced the risks of death in both DFS and OS (*p* < 0.001). 

Subgroup analyses for the patients in the low, single high, and both high groups were performed to explore the effect of endocrine therapy based on the HR expression level of the breast tumors (Figure 7). The DFS and OS improved in the patients in all the three HR groups who received endocrine therapy. Treatment effectiveness was the highest for the patients in the dual high group, followed by those in the single high group (*p* < 0.05). In the low HR expression level group, statistical significance for endocrine treatment could not be confirmed, but the patients receiving endocrine therapy showed favorable results (DFS, *p* = 0.054; OS, *p* = 0.165).

## 4. Discussion

The HR expression level could serve as a prognostic marker for breast cancer. This study reconfirmed that HR-positive breast cancer with low levels of both ER and PgR expression had clinicopathologic patterns similar to HR-negative breast cancer. We also found that endocrine therapy was strongly considered in all the patients with HR-positive breast cancer, regardless of the HR expression level of the breast tumor. 

According to the St. Gallen consensus [14,23], breast cancer subtypes such as luminal A or B, HER2-positive, and triple-negative are classified based on the positivity of ER or PgR expression, HER2 overexpression, and Ki-67 expression related to cell proliferation. Among the luminal subtype, breast cancer only with a positive HR, no HER2 overexpression, and low Ki-67 expression can be included in the luminal A subtype. Furthermore, breast cancer with the luminal B subtype showed clinical characteristics, such as a high risk of early metastasis, and a disseminated metastatic pattern related to a worse prognosis than that in the luminal A subtype [22,23]. In this study, the group comprising patients with breast cancers with high expression of a single receptor showed the worst survival outcomes of the patients, among the different groups of HR expression level. 

Results of long-term follow-up for breast cancer, according to the positivity of HR, have been published earlier [24,25,26]. For patients with HR-negative cancer, the prognosis deteriorated rapidly for the first 2–3 years after treatment, and plateaued thereafter. After five years, a higher recurrence rate was observed for HR-positive cancers than for HR-negative cancers, suggesting a different clinical course of HR-positive versus HR-negative breast cancers with long-term patient follow-up. In our study, breast cancers with high expression of a single HR, which might have been categorized as HR-positive cancers in previous studies, showed a continuous increase in the recurrence rates during long-term patient follow-up. In addition, a 5–10-year duration of endocrine therapy could affect the patients’ prognosis, but we could not analyze the detailed duration of treatment in this study. Additional validation is required by reflecting group classification and actual treatment. 

As PgR is expressed according to ER activity [27,28], there are several controversial theories about the expression of PgR alone without ER-positivity [29,30,31]. For data accuracy, ASCO recommended that IHC staining should be performed using a validated method in an appropriate facility and examined by a professional pathologist [32]. In this study, we only reviewed results of cancers with ER and PgR expression from permanent pathology reports, without re-testing for HR. However, very few cases of ER−/PR+ breast cancers have been annually reported, which was also observed in other studies [19,33]. Further studies should focus on breast cancers with discordantly expressed ER and PgR. 

There have been several attempts to use the level of HR expression along with other predictive factors of breast cancer. In 2012, Dutch researchers found that a combination of the levels of ER/PgR expression and 70-gene signature (MammaPrint™, Agendia BV, Amsterdam; cleared by the FDA for risk assessment) was associated with 10-year breast cancer-specific patient survival [34]. A total of 121 ER-positive patients with breast cancer treated with mono-tamoxifen were selected. They were categorized as either highly or incompletely responsive to endocrine therapy based on ER/PgR expression of the tumor being ≥50%. After the ER/PgR expression level was adjusted for the 70-gene signature using multivariate Cox regression analysis, patients in the highly endocrine-responsive category were associated with good breast cancer-specific survival (hazard ratio = 0.16). The combination of the ER/PgR expression level of the tumor and a multigene assay might provide an improved prediction of the survival outcome of the patient. In 2017, Truin et al. connected the level of ER/PgR expression to histologic types of breast cancer to evaluate chemosensitivity [35]. A total of 26,339 ER/PgR-positive subjects diagnosed with invasive ductal carcinoma (IDC) or invasive lobular carcinoma (ILC) were analyzed. The positivity of ER and PgR was 10% or higher according to the Dutch guidelines. For the subjects, a semi-quantitative classification was repeated using the tumor ER expression level: 10–69%, 70–89%, and >90%. The results showed that most of the patients with ER-positive breast cancer expressed more than 90% of ER, and there was no significant difference in the ER expression level between IDC and ILC. Further studies of HR expression level alone as well as in combination with other predictive factors must be considered.

The current study has several advantages. First, we analyzed a larger number of patients with breast cancer expressing a low level of either ER or PgR than that assessed by previous studies. Second, we reflected the long-term time factor into the survival analysis of the patients; clinical features and progress of cancer were followed for over 20 years. Third, this study is one of the few studies to investigate a subtype of breast cancer where only one receptor is highly expressed, commonly considered typical HR-positive breast cancer. The need for endocrine therapy for the patient traditionally depends on the status of ER expression of the tumor. However, it is known that ER signaling influences the process of PgR gene expression [29,36,37]. In this study, the HR group was newly organized by the combination of ER and PgR expression levels and was given significance as a prognostic factor for breast cancer. Lastly, informatics in the form of electronic medical record data extraction as well as general data sources from medical records or accumulated databases were used throughout the data collection process.

However, there are several limitations of the present study. First, the proportion of patients with breast cancers expressing a low level of both ER and PgR was minute compared to the total number of patients. The distribution of breast cancer has a biphasic pattern according to the level of HR expression [38]. Therefore, it is natural that most of the patients were included in the negative group and in the both high group. Second, this study cannot represent the general population because of the retrospective nature of the collected data from one institution. For the same reason, it is hard to realistically reflect the compliance of patients with long-term adjuvant therapies and to investigate an individual’s details in the process of treatment. Finally, antibody types for the detection of ER and PgR were inconsistent in all the patients during the long study period. The types of antibodies used for IHC testing for ER and PgR have changed over time as antibodies for more effective detection have been developed. 

## 5. Conclusions

HR expression levels have prognostic value for patients with breast cancer. The PgR as well as ER expression level of the tumor should be considered, as clinicopathological characteristics and patient treatment may vary depending on the HR expression levels of the tumor. This study demonstrated the long-term relapse characteristics of single HR-positive breast cancer. Endocrine therapy has favorable effects, even in patients with breast cancer expressing low levels of HR.

## Figures and Tables

**Figure 1 cancers-13-05007-f001:**
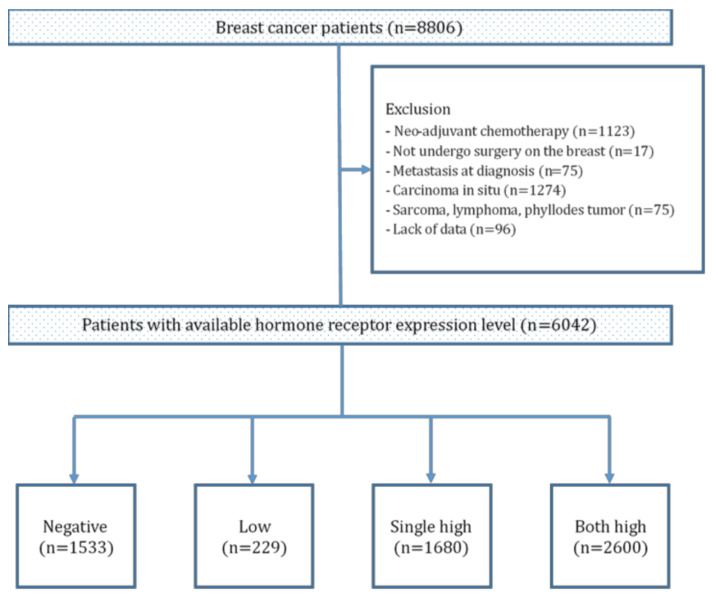
Schematic representation of the process of patient selection.

**Figure 2 cancers-13-05007-f002:**
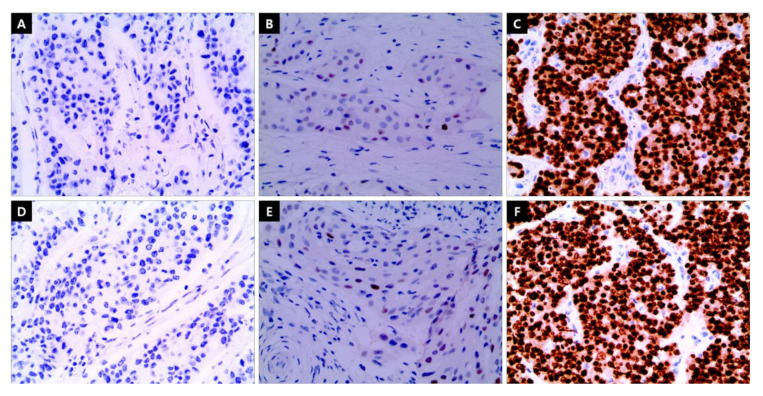
Immunohistochemistry staining of estrogen receptor and progesterone receptor. (**A**) Negative: <1% (**B**), low: 1–10%, and (**C**) high: >10% expression levels of estrogen receptor. (**D**) Negative: <1% (**E**), low: 1–20%, and (**F**) high: >20% expression levels of progesterone receptor. (×400) Scale bar or magnification.

**Figure 3 cancers-13-05007-f003:**
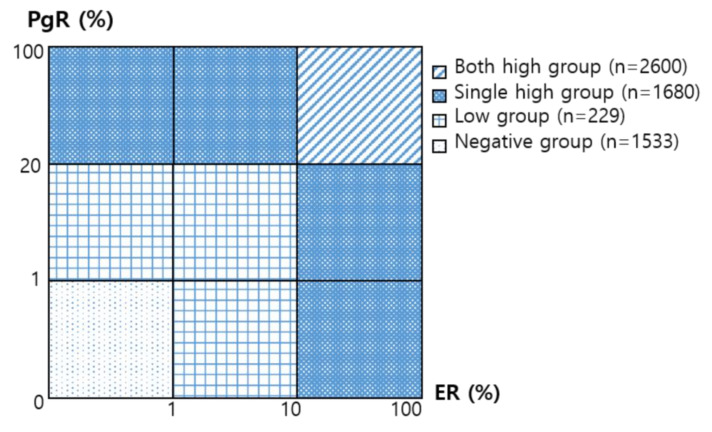
The definition of hormone receptor group, according to the combination of estrogen receptor (ER) and progesterone receptor (PgR) expression level. The negative group: ER/PgR < 1%/<1%. The low group: ER/PgR < 1%/1–20%, 1–10%/<1%, and 1–10%/1–20%. The single high group: ER/PgR 0–10%/>20% and >10%/0–20%. The both high group: ER/PgR >10%/>20%.

**Figure 4 cancers-13-05007-f004:**
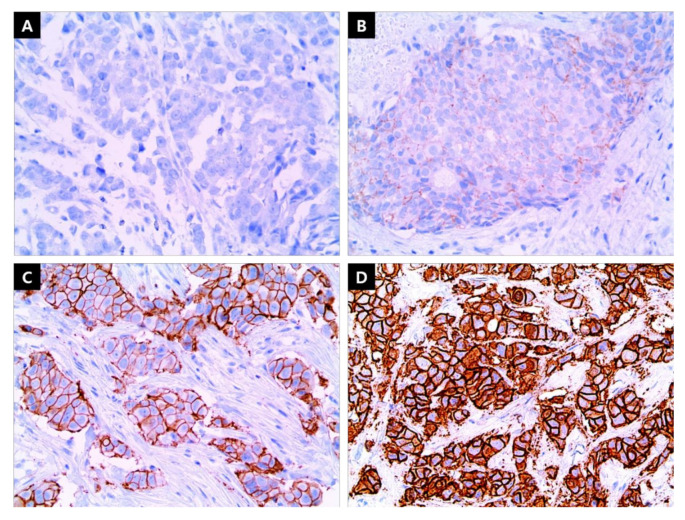
Immunohistochemistry (IHC) staining for human epidermal growth factor receptor 2 (HER2) expression. (**A**) Negative, (**B**) HER2 IHC 1+, (**C**) HER2 IHC 2+, and (**D**) HER2 IHC 3+. (×400) Scale bar or magnification.

**Figure 5 cancers-13-05007-f005:**
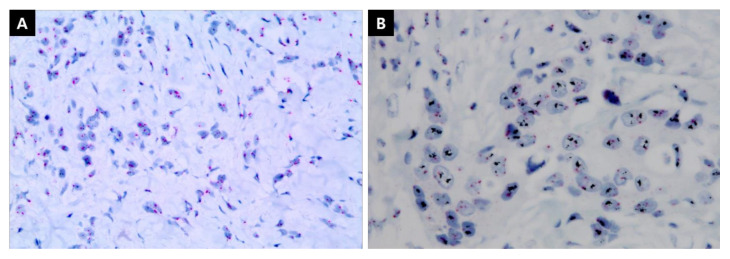
Silver in situ hybridization (SISH) evaluation to determine HER2 overexpression in breast cancer. (**A**) SISH-negative and (**B**) SISH-positive. (×400) Scale bar or magnification.

**Figure 6 cancers-13-05007-f006:**
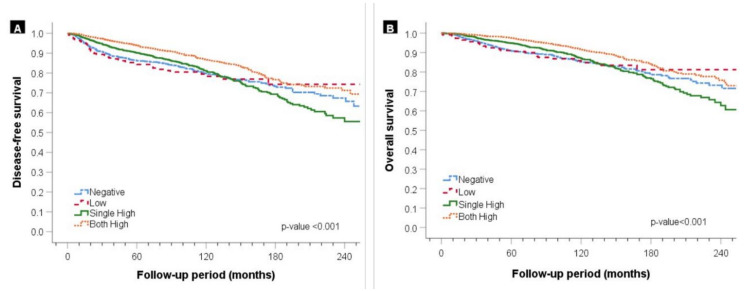
Kaplan–Meier survival analysis according to the expression level of the hormone receptor. (**A**) Disease-free survival and (**B**) overall survival.

**Figure 7 cancers-13-05007-f007:**
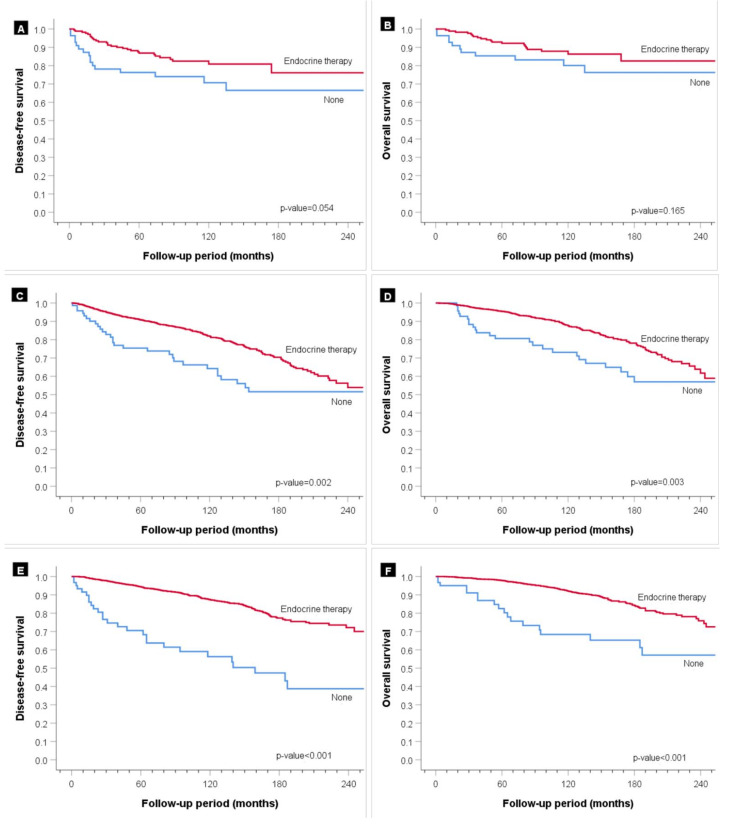
Survival analysis of the patients in the three groups with hormone receptor-positive status according to endocrine therapy. (**A**) Disease-free survival (DFS) and (**B**) overall survival (OS) for the low group, (**C**) DFS and (**D**) OS for the single high group, and (**E**) DFS and (**F**) OS for the both high group.

**Table 1 cancers-13-05007-t001:** Baseline characteristics of patients classified according to the groups of hormone receptor levels.

	Group of ER/PgR Level Combination				
	Negative *n* = 1533 (%)	Low *n* = 229 (%)	Single High *n* = 1680 (%)	Both High *n* = 2600 (%)	*p*-Value
**Patient Age** (**Years**)					<0.001
<50	709 (46.2)	110 (48.0)	691 (41.1)	1595 (61.3)	
≥50	824 (53.8)	119 (52.0)	989 (58.9)	1005 (38.7)	
**BMI** (**kg/m^2^**)					<0.001
<23	746 (48.7)	104 (45.4)	815 (48.5)	1362 (54.4)	
≥23	739 (48.2)	120 (52.4)	836 (49.8)	1210 (46.5)	
Unknown	48 (3.1)	5 (2.2)	29 (1.7)	28 (1.1)	
**Tumor Histology**					<0.001
Ductal	1366 (89.1)	208 (90.8)	1463 (87.1)	2261 (87.0)	
Lobular	15 (1.0)	5 (2.2)	60 (3.6)	132 (5.0)	
Other	152 (9.9)	16 (7.0)	157 (9.3)	207 (8.0)	
**Tumor Stage**					<0.001
T1	857 (55.9)	132 (57.6)	1129 (67.2)	1810 (69.6)	
T2	638 (41.6)	90 (39.3)	522 (31.1)	764 (29.4)	
T3, T4	38 (2.5)	7 (3.1)	29 (1.7)	26 (1.0)	
**Node Stage**					<0.001
N0	1150 (75.0)	163 (71.2)	1141 (67.9)	1782 (68.5)	
N1	272 (17.7)	39 (17.0)	383 (22.8)	602 (23.2)	
N2	70 (4.6)	17 (7.4)	92 (5.5)	153 (5.9)	
N3	41 (2.7)	10 (4.4)	64 (3.8)	63 (2.4)	
**Histologic Grade**					<0.001
I	84 (5.4)	17 (7.5)	447 (26.6)	919 (35.3)	
II	530 (34.6)	110 (48.0)	876 (52.1)	1297 (49.9)	
III	909 (59.3)	93 (40.6)	300 (17.9)	290 (11.2)	
Unknown	10 (0.7)	9 (3.9)	57 (3.4)	94 (3.6)	
**HER2 Overexpression**					<0.001
Negative	1013 (66.1)	139 (60.7)	1307 (77.8)	2191 (84.3)	
Positive	414 (27.0)	72 (31.4)	252 (15.0)	225 (8.7)	
Undetermined	106 (6.9)	18 (7.9)	121 (7.2)	184 (7.1)	
**Breast Surgery** (**Mastectomy**)					0.133
Partial	689 (44.9)	93 (40.6)	801 (47.7)	1215 (46.7)	
Total	844 (55.1)	136 (59.4)	879 (52.3)	1385 (53.3)	
**Radiation Therapy**					0.093
None	710 (46.3)	110 (48.0)	713 (42.4)	1139 (43.8)	
Performed	823 (53.7)	119 (52.0)	967 (57.6)	1461 (56.2)	
**Adjuvant Chemotherapy**					<0.001
None	314 (20.5)	41 (17.9)	624 (37.1)	1042 (40.1)	
Performed	1219 (79.5)	188 (82.1)	1056 (62.9)	1558 (59.9)	
**Endocrine Therapy**					<0.001
None	1480 (96.5)	56 (24.5)	72 (4.3)	66 (2.5)	
Performed	53 (3.5)	173 (75.5)	1608 (95.7)	2534 (97.5)	

ER: estrogen receptor, PgR: progesterone receptor, BMI: body mass index, HER2: human epidermal growth factor receptor 2.

**Table 2 cancers-13-05007-t002:** Multivariate Cox regression analysis of disease-free survival and overall survival by baseline characteristics.

	Disease-Free Survival		Overall Survival	
	Hazard Ratio (95% CI)	*p*-Value	Hazard Ratio (95% CI)	*p*-Value
**Patient Age** (**Years**)		0.486		<0.001
<50	1		1	
≥50	1.05 (0.92–1.20)		1.48 (1.25–1.74)	
**BMI** (**kg/m^2^**)		<0.001		<0.001
<23	1		1	
≥23, Unknown	1.41 (1.24–1.61)		1.43 (1.22–1.68)	
**Tumor Size**		<0.001		<0.001
≤2 cm	1		1	
>2 cm	1.73 (1.50–1.99)		1.65 (1.40–1.95)	
**Node Metastasis**		<0.001		<0.001
Negative	1		1	
Positive	1.97 (1.69–2.29)		1.98 (1.65–2.37)	
**Histologic Grade**		0.193		0.682
I, II	1		1	
III, Unknown	1.10 (0.95–1.28)		1.04 (0.87–1.23)	
**Group of ER/PR Level**				
Negative	0.71 (0.51–0.99)	0.046	1.04 (0.73–1.49)	0.810
Low	1.25 (0.89–1.75)	0.204	1.27 (0.85–1.90)	0.238
Single high	1.33 (1.14–1.56)	<0.001	1.42 (1.18–1.72)	<0.001
Both high	1		1	
**HER2 Overexpression**				
Negative	1		1	
Positive	0.81 (0.67–0.97)	0.021	0.92 (0.74–1.14)	0.463
Undetermined	1.98 (1.62–2.42)	<0.001	1.48 (1.18–1.86)	0.001
**Breast Surgery** (**Mastectomy**)		<0.001		<0.001
Partial	1		1	
Total	1.89 (1.53–2.33)		1.97 (1.53–2.54)	
**Radiation Therapy**		0.640		0.239
None	1		1	
Performed	0.96 (0.79–1.15)		1.14 (0.92–1.41)	
**Adjuvant Chemotherapy**		0.002		<0.001
None	1		1	
Performed	0.77 (0.65–0.91)		0.64 (0.53–0.79)	
**Endocrine Therapy**		<0.001		0.001
None	1		1	
Performed	0.41 (0.30–0.55)		0.60 (0.44–0.82)	

CI: confidence interval, BMI: body mass index, ER: estrogen receptor, PgR: progesterone receptor, HER2: human epidermal growth factor receptor 2.

## Data Availability

The data presented in this study are available on request from the corresponding author. The data are not publicly available, because the information contained in them could compromise the privacy of the research participants.

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
