# Peer review of "Level of Combined Estrogen and Progesterone Receptor Expression Determines the Eligibility for Adjuvant Endocrine Therapy in Breast Cancer Patients"

_cancers, 2021, doi:10.3390/cancers13195007_

Round 1

Reviewer 1 Report

This manuscript describes outcomes data from a large series of breast cancer patients in Korea. This type of study has been reported many times and over several decades. However, with changes in the diagnosis and management of breast cancer, and the knowledge that is gained from different populations worldwide, I feel that manuscripts such as the present one still provide important knowledge. The long follow-up of patients included in this cohort is another positive aspect of the study.

In general, the manuscript is well-written, although multiple minor English language corrections will be required. The tables are clearly presented, however I feel that the figures are too small and that font sizes in the figures should be increased (see below). Once figure sizes are increased, the thickness of lines in the figures may also need to be increased, to improve the visibility of results. Referencing is appropriate, however a number of references are quite old and should be updated. Due to the type and scope of data, I feel that the manuscript overall is worthy of publication, however I would suggest that expansion of the analysis and discussion topics of the manuscript would enhance the value of the work.

  1. Some of the references cited in this manuscript are quite old and there are more recent references that could be used either in addition to or instead of those listed. This is particularly true for the references cited in the first paragraph of the Introduction.
  2. There have been quite a few studies investigating why ER+/PR- and ER-/PR+ cancers occur. Although it is not necessary to delve too deeply into the scientific details of those studies, the findings are relevant to the present work as they begin to address why breast cancers which only express one of these receptors may behave differently to those that express high levels of both receptors. I feel that this information should be added to the manuscript.
  3. The origin of the patient (details) that are included in the study are not clear. Although it is stated that the Severance Hospital Yonsai University Health System approved the study, the source of the patient data should be added (e.g. were these all patients whose surgery was performed at this hospital).
  4. Please add a reference to line 107 (AJCC).
  5. In lines 241-242, the authors “hypothesise that the group of single receptor-high expression (sic) could show clinically similar behaviour to the luminal B subtype”. I don’t understand the basis or the relevance of this statement. If laboratories are delineating cancers that fit the criteria of luminal B, then these should be listed. Although it is possible that this group contains cancers that could be classified as luminal B, it is unlikely that all are luminal B tumours. I would suggest removing this speculative statement unless further evidence and/or justification can be added.
  6. The authors identify a strength of their study to be their separate analysis of breast cancers where just one of ER or PR are highly expressed. I would agree with this statement, however I feel that further information could be added including details of the number of tumours that expressed ER (only) at high levels and the number expressing only PR at high levels. It may also be worth referencing previous research examining the underlying biology or diagnostic pathology technical issues (in the case of PR+/ER- cases) associated with these tumours.
  7. Previous studies have highlighted the continuing relapse of successfully treated ER+/PR+ breast cancers, in contrast to the higher early relapse/metastasis of ER-/PR- tumours, which then plateaus. How do the authors feel that these previous findings are reflected in their data?
  8. Please increase the overall size and the font size of figures. It may also be necessary to increase the thickness of lines in the figures in order to improve their visibility.

Author Response

Q1-1. Some of the references cited in this manuscript are quite old and there are more recent references that could be used either in addition to or instead of those listed. This is particularly true for the references cited in the first paragraph of the Introduction.

  • We reflect the latest reference within 3 years in the first paragraph of the Introduction (page 2).

Q1-2. There have been quite a few studies investigating why ER+/PR- and ER-/PR+ cancers occur. Although it is not necessary to delve too deeply into the scientific details of those studies, the findings are relevant to the present work as they begin to address why breast cancers which only express one of these receptors may behave differently to those that express high levels of both receptors. I feel that this information should be added to the manuscript.

  • We added more information of breast cancers which only express one of ER or PR, wrote sentences in Introduction and indicated references (page 2, line 78-81).

Q1-3. The origin of the patient (details) that are included in the study are not clear. Although it is stated that the Severance Hospital Yonsai University Health System approved the study, the source of the patient data should be added (e.g. were these all patients whose surgery was performed at this hospital).

  • We added the information of the hospital (page 3, line 98).

Q1-4. Please add a reference to line 107 (AJCC).

  • We added the reference of AJCC 7th edition (page 3, line 111)

Q1-5. In lines 241-242, the authors “hypothesise that the group of single receptor-high expression (sic) could show clinically similar behaviour to the luminal B subtype”. I don’t understand the basis or the relevance of this statement. If laboratories are delineating cancers that fit the criteria of luminal B, then these should be listed. Although it is possible that this group contains cancers that could be classified as luminal B, it is unlikely that all are luminal B tumours. I would suggest removing this speculative statement unless further evidence and/or justification can be added.

  • We agree with your comment. There is no further evidence, so we deleted the sentence.

Q1-6. The authors identify a strength of their study to be their separate analysis of breast cancers where just one of ER or PR are highly expressed. I would agree with this statement, however I feel that further information could be added including details of the number of tumours that expressed ER (only) at high levels and the number expressing only PR at high levels. It may also be worth referencing previous research examining the underlying biology or diagnostic pathology technical issues (in the case of PR+/ER- cases) associated with these tumours.

  • We did additional analysis, and there was no significant difference in survival outcome between the two groups. We added sentences in Result (page 9, line 219-222) and Discussion (page 10, line 277-285).

Q1-7. Previous studies have highlighted the continuing relapse of successfully treated ER+/PR+ breast cancers, in contrast to the higher early relapse/metastasis of ER-/PR- tumours, which then plateaus. How do the authors feel that these previous findings are reflected in their data.

  • We discussed this review in Discussion (page 10, line 259-269)

Q1-8. Please increase the overall size and the font size of figures. It may also be necessary to increase the thickness of lines in the figures in order to improve their visibility.

  • We increased the size of figures to be easy to see. We confirmed sufficient high resolution of pictures with 350 x 350 DPI.

Reviewer 2 Report

Overall it is a nice MS. However, there is a room to improve it.

Critical comments:

  1. Simple summary is not clear. Authors may rewrite it in simplistic manner.
  2. Please provide representative IHC pictures of different percentages of expressions of ER and PgR
  3. Please also provide for HER2 IHC and FISH pictures
  4. In figure 2, authors may provide better/detail legend
  5. In the single high group, is there any difference in terms of DFS or OS between ER vs PgR expression? Please mentioned clearly in the result section and discuss it appropriately in the discussion section
  6. It will be better to make discussion little shorter

Minor comment:

Instead of HoR, authors may write HR (for hormone receptors)

Author Response

Q2-1. Simple summary is not clear. Authors may rewrite it in simplistic manner.

  • We made Simple summary more simple and clear.

Q2-2. Please provide representative IHC pictures of different percentages of expressions of ER and PgR.

  • We provided the IHC pictures of ER and PgR with negative and expression of low and high levels in Figure 2.
  •  

Q2-3. Please also provide for HER2 IHC and FISH pictures.

  • We provided the IHC pictures of HER2 negative, +, ++, +++ status in Figure 4. Also, we provided positive and negative of SISH pictures in Figure 5.

Q2-4. In figure 2, authors may provide better/detail legend

  • The previous figure 2 was changed to Figure 3. We described the details in the level of each hormone receptor group.

Q2-5. In the single high group, is there any difference in terms of DFS or OS between ER vs PgR expression? Please mentioned clearly in the result section and discuss it appropriately in the discussion.

  • We did additional analysis, and there was no significant difference in survival outcome between the two groups. We added sentences in Result (page 9, line 219-222) and Discussion (page 10, line 277-285).

Q2-6. It will be better to make discussion little.

  • We agreed with the comment. We reviewed the whole discussion and figured out that a previous paragraph discussed on low level of each estrogen or progesterone receptors already mentioned Introduction. We removed that paragraph and also clarified other things.
  •  

Q2-7. Instead of HoR, authors may write HR (for hormone receptors)

  • When we wrote HR, there were no other words or phrases that could cause confusion (e.g. hazard ratio), so we replaced HoR with HR.

Round 2

Reviewer 1 Report

Then authors have addressed many of the reviewers’ comments and added or amended several aspects of the manuscript text and figures. For these reasons, I feel that the manuscript is acceptable for publication. However, English language remains poor throughout the text and this detracts from the overall content of the manuscript. I recommend that the authors work with an English language editor who has experience with scientific terminology and in particular the language of medical research literature.

  1. I can see that the authors have attached an English language editing certificate, however there are significant English language errors that remain in the manuscript. The most obvious ones occur in relation to hormone receptor levels, breast cancers, cancer patients and outcomes/survival. For the purposes of this manuscript, “patients” do not express hormone receptors (HRs), their breast tumours express HRs. “Patients” have survival outcomes, not breast tumours, etc. The confusion in this basic English language usage is quite distracting and should be corrected throughout the document. I have provided suggested corrections for the Simple Summary below, however all parts of the manuscript require careful amendment.

Patients whose breast cancers express low levels of hormone receptor (HR) could be eligible for adjuvant endocrine therapy, but limited data are available to support this. Our retrospective study investigated the characteristics and survival of 6,042 breast cancer patients according to four HR groups of combined estrogen and progesterone receptor expression. HR expression levels were prognostic for recurrence and death of breast cancer patients. Patients whose tumors expressed high levels of just a single HR had the worst survival outcomes, and their risk of death continuously increased even after 10-years’ follow-up. Endocrine therapy had a significant benefit for those whose tumors expressed high HR levels and a favorable tendency for patients with tumors expressing low HR levels. We confirmed the value of HR expression level as a prognostic factor and the possible benefit of endocrine therapy in cases where breast tumors expressed low HR levels.

  1. A number of phrases repeated throughout the manuscript are not English, for example, “single receptor-high expression”. These should all be replaced with phrases that are grammatically correct and consistent with other manuscripts published in journals with the appropriate specialisation (noting that HR expression refers to breast tumours and not patients). There are other sentences that do not make sense in their current form and the authors would have to work with English language editors to communicate their intended meaning. An example of this is “Breast cancers with higher than 10% of ER and 20% of PgR can show an optimized tendency for typical HR-positive breast cancers”. An English language editor without knowledge of the breast cancer literature could not work out what the authors were trying to communicate here, and may not recognise that the sentence does not make sense (scientifically) in its current form.

Author Response

1. I can see that the authors have attached an English language editing certificate, however there are significant English language errors that remain in the manuscript. The most obvious ones occur in relation to hormone receptor levels, breast cancers, cancer patients and outcomes/survival. For the purposes of this manuscript, “patients” do not express hormone receptors (HRs), their breast tumours express HRs. “Patients” have survival outcomes, not breast tumours, etc. The confusion in this basic English language usage is quite distracting and should be corrected throughout the document. I have provided suggested corrections for the Simple Summary below, however all parts of the manuscript require careful amendment.

-> I reflect your comments to my whole manuscript. I also referred to the style of your “simple summary”.

2. A number of phrases repeated throughout the manuscript are not English, for example, “single receptor-high expression”. These should all be replaced with phrases that are grammatically correct and consistent with other manuscripts published in journals with the appropriate specialisation (noting that HR expression refers to breast tumours and not patients). There are other sentences that do not make sense in their current form and the authors would have to work with English language editors to communicate their intended meaning. An example of this is “Breast cancers with higher than 10% of ER and 20% of PgR can show an optimized tendency for typical HR-positive breast cancers”. An English language editor without knowledge of the breast cancer literature could not work out what the authors were trying to communicate here, and may not recognize that the sentence does not make sense (scientifically) in its current form.

->  The terms related to receptor expression in the manuscript have also been modified to more natural expressions.